# Two-dimensional Kβ-Kα fluorescence spectrum by nonlinear resonant inelastic X-ray scattering

Kenji Tamasaku [1,2] ✉, Munetaka Taguchi [3] ✉, Ichiro Inoue[1], Taito Osaka[1], Yuichi Inubushi[1,2], Makina Yabashi [1,2] & Tetsuya Ishikawa[1]

High sensitivity of the Kβ fluorescence spectrum to electronic state is widely used to investigate spin and oxidation state of first-row transition-metal compounds. However, the complex electronic structure results in overlapping spectral features, and the interpretation may be hampered by ambiguity in resolving the spectrum into components representing different electronic states. Here, we tackle this difficulty with a nonlinear resonant inelastic X-ray scattering (RIXS) scheme, where we leverage sequential two-photon absorption to realize an inverse process of the Kβ emission, and measure the successive Kα emission. The nonlinear RIXS reveals two-dimensional (2D) Kβ-Kα fluorescence spectrum of copper metal, leading to better understanding of the spectral feature. We isolate $3d$-related satellite peaks in the 2D spectrum, and find good agreement with our multiplet ligand field calculation. Our work not only advances the fluorescence spectroscopy, but opens the door to extend RIXS into the nonlinear regime.

X-ray fluorescence spectroscopy has been a standard technique for non-destructive element analysis due to the one-to-one relationship between elements and emission photon energies. Recent investigation[1–3] on the spectral shape has revealed a close correlation of the Kβ fluorescence spectrum to the electronic states, such as the spin and oxidation state of first-row transition-metal compounds. Now, X-ray fluorescence spectroscopy becomes a powerful tool for investigating even biological samples, e.g., the oxidation state of Mn in the oxygen-evolving complex of photosystem II[4–7], in addition to the various field of physics and chemistry[1–3]. The Kβ fluorescence is a radiative transition from a 1$s$ core-hole state (1$s^1$) to a 3$p$ core-hole state (3$p^5$) and does not explicitly involve a 3$d$ state. It is an exchange interaction between 3$d$ and 3$p$ electrons that reflects the 3$d$ electronic state in the Kβ fluorescence[8]. At the same time, the exchange and other interactions split the electronic states and result in heavily overlapping spectral features as we will see later. This could make it difficult to extract the key spectral components related to the electronic states of interest and hamper the in-depth discussion.

A good playground for the fluorescence spectrum is copper metal, which is a simple system but has been investigated for decades both experimentally and theoretically[9–15]. There are two approaches to analyzing the fluorescence spectrum. As for the Kβ spectrum, it can be analytically decomposed into five Lorentzians[10]. This phenomenological analysis may have a problem that each component is not necessarily connected to a particular electronic state. The Kβ spectrum can also be well reproduced as a sum of theoretical spectra calculated for possible electronic states[11,12]. However, different theoretical models can fit the spectrum, and thus the theoretical fitting may not discriminate subtle changes or differences in the electronic state. In order to perform more deterministic analysis, we need to get more spectroscopic information from the fluorescence process.

In this article, we discuss the expansion of the one-dimensional fluorescence spectrum to two dimensions. We propose a fusion of nonlinear absorption and resonant inelastic X-ray scattering (RIXS) and demonstrate our idea on copper metal. Our analysis of the 2D fluorescence spectrum resolves 3$d$-related satellite peaks, which are

[1]RIKEN SPring-8 Center, 1-1-1 Kouto, Sayo-cho, Sayo-gun, Hyogo 679-5148, Japan. [2]Japan Synchrotron Radiation Research Institute, 1-1-1 Kouto, Sayo-gun, Hyogo 679-5198, Japan. [3]Toshiba Nanoanalysis Corporation, 8 Shinsugita-cho, Isogo-ku, Yokohama, Kanagawa 235-8522, Japan. ✉e-mail: tamasaku@riken.jp; munetaka.taguchi@nanoanalysis.co.jp

hidden by a strong main peak in the conventional Kβ fluorescence spectrum. We also present multiplet ligand calculations and interpret the experimentally resolved satellites based on the theoretical prediction.

## Results

### Nonlinear RIXS scheme

Our idea to extract more information from the Kβ fluorescence process is to expand the one-dimensional spectrum to two dimensional by adding a second axis. The spectral components, which overlap in the Kβ fluorescence spectrum, may separate along the second axis direction. Suppose we could use an absorption process from $3p^5$ to $1s^1$. This is an inverse process of the Kβ fluorescence and should be governed by the same transition matrix elements. After this absorption process, the resultant $1s^1$ state might decay by emitting the Kα fluorescence, i.e., a $1s^1 \rightarrow 2p^5$ radiative transition. Consequently, we could have two tuneable parameters, the photon energies for the absorption and emission processes. This absorption-emission process is to be treated in a similar manner to RIXS[16–18] because the absorption is resonant. If we had measured the Kβ emission, the process should be described as resonance fluorescence[19].

The biggest hurdle to realize our idea is the fact that all core levels are occupied in the ground state, and consequently, the $3p^5 \rightarrow 1s^1$ absorption process is strictly forbidden by the Pauli principle. The key is sequential two-photon absorption (Fig. 1a). In this nonlinear process,

the first photon creates a $3p$ core hole, and opens a new resonant absorption channel from $3p^5$ to $1s^1$ within a very short period of the $3p$ core-hole lifetime. The second photon can excite a $1s$ core electron to the $3p$ core hole if the photon energy is carefully tuned to the resonance. We note that the sequential absorption on the core-to-core resonance was already reported in the soft-X-ray region[20,21], however, has not been combined with the X-ray emission spectroscopy.

As we use the nonlinear process and RIXS, we refer to the above process as $3p^5 \rightarrow 1s^1 \rightarrow 2p^5$ nonlinear RIXS. An inverse process, i.e., $2p^5 \rightarrow 1s^1 \rightarrow 3p^5$ nonlinear RIXS, is also possible but may be less efficient due to the shorter lifetime for the $2p^5$ state than $3p^5$, and an order of magnitude lower Kβ yield than Kα. We label the excited state after the first absorption as $i$, the intermediate state after the second absorption as $n$, the final state after the emission as $f$, and the ground state as $g$ (Fig. 1a). The excited state $i$ may include multiple hole state due to a shake-off process and cascade of the Auger decay. The shake-off process can create two-hole states having one $3p$ hole and an additional hole in $3s$, $3p$, or $3d$ subshell[22]. The pump photon energy higher than the $L_1$ absorption edge may create a $2s$, $2p$, or $3s$ hole, and initiate the Auger cascade, during which multi-hole state with at least one $3p$ hole may appear. Among various multi-hole states possible, however, a $3p^5 3d^9$ state is found important to interpret the experimental result, as we will see later.

First, we formulate the nonlinear RIXS process. We exclude the Auger cascade from the formulation for simplicity. The absorption of the first photon can be treated as an independent process from the subsequence because a promoted high-energy photo-electron hardly interacts with the rest system. Therefore, this part can be described by rate equations[23]. Below, we will omit the electrons in the high-energy continuum from the configuration expression, e.g., $|i\rangle = |3p^5\rangle$ instead of $|i\rangle = |3p^5\varepsilon\rangle$, unless we need to indicate it explicitly. On the other hand, the absorption of the second photon is resonant, and thus the $i \rightarrow n$ absorption and the $n \rightarrow f$ emission should be described as a coherent second-order process like the conventional RIXS. Combining all processes, the differential cross-section for the nonlinear RIXS may be expressed as:

$$\frac{d\sigma^{(2)}_{\mathrm{RIXS}}\left(\omega_{\mathrm{p}},\omega_{\mathrm{e}}\right)}{d\Omega_{\mathrm{e}}} = \sum_i \sigma_{gi}\tau_i \int \frac{d^2\sigma_{\mathrm{RIXS}}\left(\omega_{\mathrm{p}},\omega_2\right)}{d\hbar\omega_2 d\Omega_{\mathrm{e}}} \Phi\left(\hbar\omega_2 - \hbar\omega_{\mathrm{e}}\right) d\hbar\omega_2,$$

(1)

where $\omega_{\mathrm{p}}$ is the pump angular frequency, $\omega_{\mathrm{e}}$ is the angular frequency at which the emission is measured, $d\Omega$ is an element of solid angle into which the photon specified by the subscript is emitted, $\sigma_{gi}$ is the $g \rightarrow i$ absorption cross-section, and $\tau_i$ is the lifetime of $i$. We introduced a normalized dimensionless instrumental function, $\Phi$, which represents the finite photon-energy resolution of the detecting system. The dimension of $\sigma^{(2)}_{\mathrm{RIXS}}$ is (length)$^4$(time), which is the same as the cross-section for two-photon absorption. The double differential scattering cross-section of the RIXS part may be written as[24]:

$$\frac{d^2\sigma_{\mathrm{RIXS}}(\omega_1,\omega_2)}{d\hbar\omega_2 d\Omega_2} = m^2 r_0^2 \left(\frac{\omega_2}{\omega_1}\right) \sum_f \left| \sum_n \frac{\omega_{fn}\omega_{ni}\langle f|\hat{r}|n\rangle\langle n|\hat{r}|i\rangle}{\hbar\omega_1 - \hbar\omega_{ni} - i\Gamma_n/2} \right|^2$$
$$\times \delta\left(\hbar\omega_1 - \hbar\omega_2 - \hbar\omega_{fi}\right),$$

(2)

by keeping only a resonant term in the Kramers–Heisenberg formula and applying the dipole approximation. Here, $\omega_1$ and $\omega_2$ are the angular frequencies of the incident and emitted photons, respectively, $m$ is the electron mass, $r_0$ is the classical radius of electron, $\Gamma_n$ is the lifetime broadening of $n$, $\omega_{ab}=(E_a-E_b)/\hbar$ with $E_x$ being the energy of the state $x$, and $\hat{r}$ is the position operator of the electron. Summation over all electrons is omitted for simplicity. We point out here that the initial

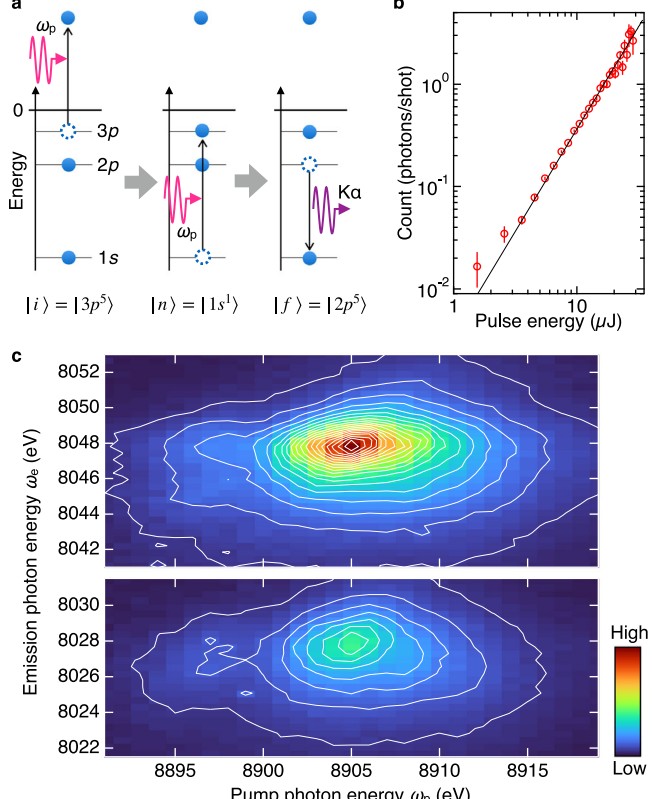

**Fig. 1 | Nonlinear RIXS of copper metal. a** Schematic nonlinear RIXS process. The system is excited by sequential two-photon absorption including the inverse process of the Kβ fluorescence, and then decays by the Kα emission. Uninvolved orbitals and electrons are omitted for simplicity. **b** The pulse-energy dependence of the 8048 eV emission count rate at a pump photon energy of 8905 eV (circles). Vertical bars indicate the standard error of the mean. Solid line represents the best fit with a quadratic function without a linear or constant term. The coefficient of the quadratic term gives $\sigma^{(2)}_{\mathrm{RIXS}}$. **c** Image plot of $\sigma^{(2)}_{\mathrm{RIXS}}$. The pump photon energy covers the whole Kβ spectral range, while the emission photon energy does the main parts around the Kα$_1$ and Kα$_2$ peaks.

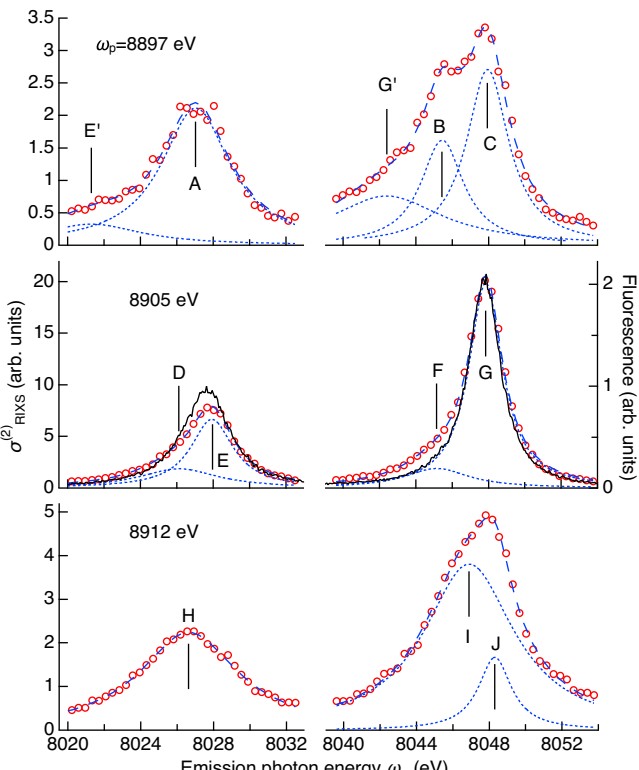

**Fig. 2 | 2D fluorescence spectrum section at constant $\omega_p$.** Circles show sections of $\sigma^{(2)}_{RIXS}$ at three characteristic $\omega_p$. The error of $\sigma^{(2)}_{RIXS}$ due to the fitting with a quadratic function (Fig. 1b) is smaller than the marker size. Solid lines in the middle panel represent the Kα fluorescence spectrum measured by one-photon excitation at 9100 eV well above the K edge. Dashed lines represent the fitting with Lorentzian(s), and each component is plotted by a dotted line.

state of the RIXS part includes many excited states with different energies, whereas the conventional RIXS starts from the ground state.

## Measurement of $\sigma^{(2)}_{RIXS}$

Since the lifetime of the $3p$ core-hole state, $\tau_i$, is on a sub-femtosecond timescale, nonlinear RIXS requires intense femtosecond X-rays. We used an X-ray free-electron laser, SACLA[25], which delivered an X-ray beam with the central photon energy at 8905 eV, corresponding to the Kβ photon energy of copper, a full-width-at-half-maximum (FWHM) bandwidth of 43 eV, and an FWHM pulse duration as short as 8 fs (ref. 26). We used a Si (111) double-crystal monochromator to select a 1.0 eV monochromatic beam, and to scan the pump-photon energy over the Kβ range, and then, focused the beam down to 0.72(H) × 0.92(V) μm² (FWHM) by a Kirkpatrick–Baez (KB) mirror[27] (see Methods and Supplementary Fig. 2 for details). The average pulse energy in front of the KB mirror was 8.0 μJ at 8905 eV, from which we estimate the peak intensity and the peak photon-flux density on the focus to be $1.5 \times 10^{17}$ W/cm² and 11 photons/Å²/fs, respectively. We measured the Kα emission spectrum from a 10-μm-thick copper foil set on the focus using a single-shot polychromator.

Figure 1b shows a typical pump pulse-energy dependence of the emission count rate measured at $\omega_e$ = 8048 eV, i.e., the peak of the Kα₁ fluorescence spectrum. The pump photon energy was tuned to the Kβ fluorescence peak at $\omega_p$ = 8905 eV (Supplementary Fig. 3). The pulse-energy dependence is well reproduced by a quadratic term only, as expected for sequential two-photon absorption. This indicates that non-resonant fluorescence by higher harmonic radiation and multi-photon processes of more than two are negligible under the present condition. The coefficient of the quadratic term is proportional to $\sigma^{(2)}_{RIXS}$. Analyzing the pulse-energy dependence of the Kα emission

counts at each pump and emission photon energies, we map out $\sigma^{(2)}_{RIXS}$ on the $\omega_p$–$\omega_e$ plane as shown in Fig. 1c.

## 2D fluorescence spectrum

The $\sigma^{(2)}_{RIXS}$ plane shows two prominent peaks at $(\omega_p, \omega_e)$=(8905 eV, 8027 eV) and (8905 eV, 8048 eV), which correspond to the peak photon energies for the Kβ and Kα₁,₂ fluorescence spectra. These two peaks are not simple Lorentzian but are accompanied by additional weak peaks. To make clear the weak structures, three characteristic sections along constant $\omega_p$ lines are shown in Fig. 2. We also plot the Kα fluorescence spectrum measured at an excitation photon energy of 9100 eV well above the K edge at 8980 eV for comparison. Among the three emission spectra, only that at $\omega_p$ = 8905 eV, i.e., the Kβ peak, is found similar to the Kα fluorescence spectrum.

Here, we employ the phenomenological approach to analyze the emission spectra. The Kα emission spectrum is much simpler than Kβ because the exchange interaction between the $3d$ and $2p$ electrons is weaker than that between $3d$ and $3p$. We may associate analytically decomposed Lorentzians with particular electronic transitions. The Kα₁ emission peak around $\omega_e$ = 8048 eV can be decomposed into two Lorentzians labeled F and G. The spectral shape of the main peak G coincides with the Kα₁ fluorescence spectrum. Component G is considered to originate predominantly from the $1s^1 \rightarrow 2p^5$ diagram transition. Comparing with the previous analysis on the Kα fluorescence spectrum[10,13], we assign the weaker peak F to a satellite due to the $1s^1 3d^9 \rightarrow 2p^5 3d^9$ transition. However, we consider that component F may contain just a part of the $1s^1 3d^9 \rightarrow 2p^5 3d^9$ transition, and the rest may be included in the main component G. The Kα₂ peak also consists of two Lorentzians D and E, which are considered to have the same origins as F and G, respectively. The branching ratio (Kα₂/Kα₁) of the nonlinear RIXS is found slightly higher than the Kα fluorescence, which may be due to the resonant excitation at a single photon energy of 8905 eV.

When $\omega_p$ was detuned from the Kβ peak, the Kα emission spectrum changed shape as expected for RIXS. We need at least three Lorentzians, G´, B, and C, to reproduce the Kα₁ emission spectrum at $\omega_p$ = 8897 eV. The broad peak labeled G´ lies near the constant energy transfer line, $\omega_p$–$\omega_e$ = 857 eV, which passes through the peak G on the two-dimensional spectrum. Therefore, we consider G´ is the tail of G. This is a characteristic of the coherent process, where the conservation of energy must be satisfied between the states $i$ and $f$, but not for the intermediate state $n$ in Eq. (2). On the other hand, the Kα₁ emission spectrum for $\omega_p$ = 8912 eV has no sign of the tail of G, and can be reproduced by two Lorentzians, I and J. The three peaks, C, G, and J, have similar central photon energies, indicating that they have the same origin, while larger energy difference among the satellite peaks, B, F, and I, implies that they might come from different electronic states. Unlike Kα₁, the Kα₂ emission spectra can be reproduced with one or two components.

## Analysis of $\sigma^{(2)}_{RIXS}$

Now, we apply the same procedure as Fig. 2 to decompose all emission spectra into Lorentzian components in order to understand the overall structure of $\sigma^{(2)}_{RIXS}$. We independently fitted the emission spectrum at each $\omega_p$ with a minimum number of Lorentzians and no constant background (Supplementary Fig. 4). One 2D spectral feature may be sliced into several one-dimensional Lorentzians. We did not impose any constraint on the fitting parameters, such as the peak photon energy, to connect a Lorentzian to its neighbors along the $\omega_p$ direction. Figure 3a and b show the decomposition result in the Kα₁ and Kα₂ regions, respectively. The components in the Kα₁ region can be classified into several clusters, while those for Kα₂ are not dissolved well as is pointed out above.

To decipher the spectral feature, we compare theoretical calculation based on Eq. (1). We used the standard multiplet ligand field

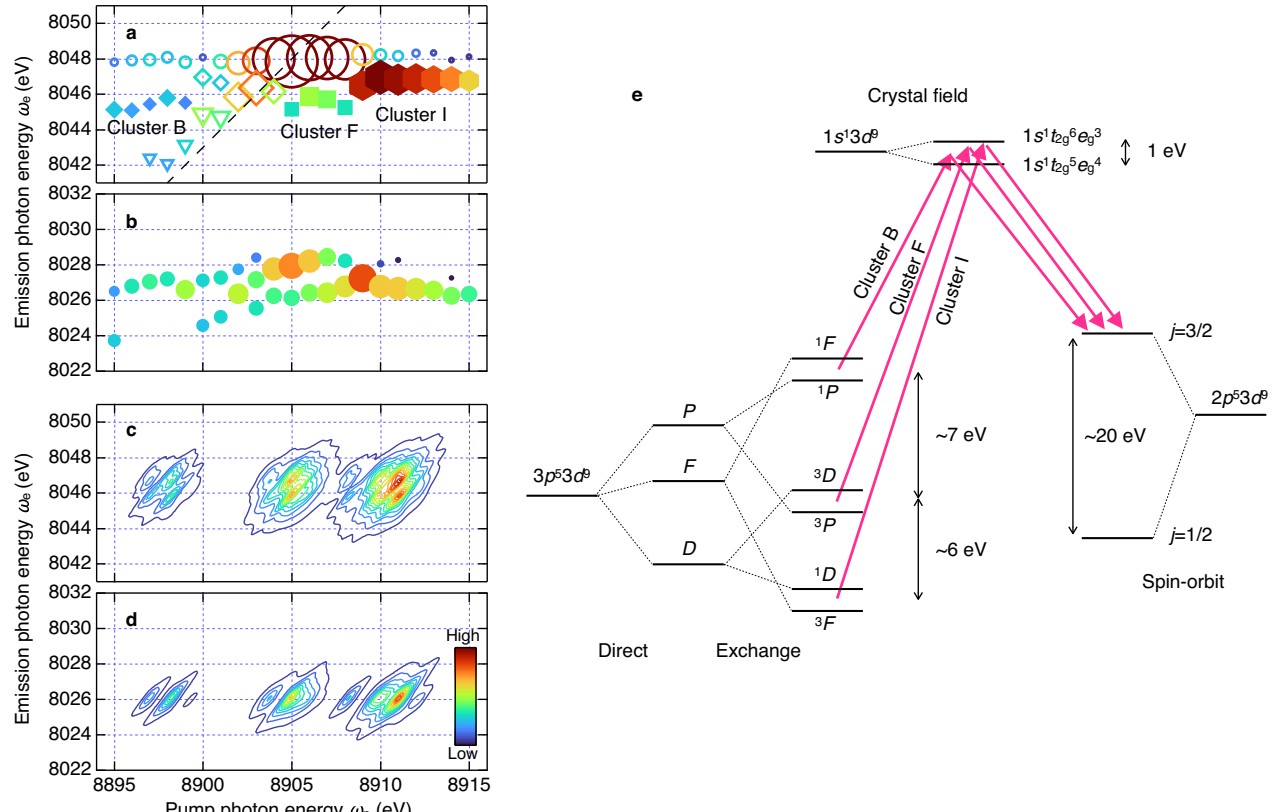

**Fig. 3 | Decomposition of Kα emission spectra, and theoretical calculation for $3p^5 3d^9$ satellite peaks. a** Decomposition of the Kα$_1$ emission spectra. Each symbol and its size represent the peak center and area of the Lorentzian, respectively. Components denoted by the same symbol are considered to have the same origin. The cluster ID for closed diamonds (B), squares (F), and hexagons (I) is taken from the corresponding peak in Fig. 2. Open circles and open diamonds are considered to be the main peak and the Kα$_1$ fluorescence from relaxed state (see text for details). Triangles are assigned to the tail of main peak. Dashed line indicates the

constant energy transfer line, $\omega_p - \omega_e = 857$ eV, which pass through the main peak. **b** Decomposition of the Kα$_2$ emission spectra. Circles indicate the center of the Lorentzian with the peak area represented by the size. **c, d** Theoretical $\sigma^{(2)}_{RIXS}$ calculated for the $3p^5 3d^9$ satellite peaks. The calculated spectrum is shifted to align with the experimental data. **e** Schematic energy-level diagram of copper metal for $|i\rangle = |3p^5 3d^9\rangle$, $|n\rangle = |1s^1 3d^9\rangle$, and $|f\rangle = |2p^5 3d^9\rangle$. The energy difference is taken from the theoretical calculation.

theory[16,17,28] with octahedral ($O_h$) crystal fields of a face-centered cubic structure of copper metal (see the "Methods" section for details). In the rest of this article, we focus on the $3p^5 3d^9$ satellites, i.e., transitions involving $|i\rangle = |3p^5 3d^9\rangle$. We ignored the $i$-dependence of $\tau_i$ in Eq. (1), and calculated the scattering cross-section of the RIXS part as $3p^5 3d^9 \varepsilon \rightarrow 1s^1 3d^9 \varepsilon \rightarrow 2p^5 3d^9 \varepsilon$ using Eq. (2), where we considered a Cu$^{2+}$ ion instead of explicitly taking into account the shake-off process[10].

Figure 3c and d show $\sigma^{(2)}_{RIXS}$ calculated for the $3p^5 3d^9$ satellites. Our simulation predicted three peaks for both the Kα$_1$ and Kα$_2$ emission. Since the $3p^5 3d^9$ configuration has two holes, the possible $LS$ terms are $^{1,3}P$, $^{1,3}D$, and $^{1,3}F$, corresponding to the orbital angular momentum of $L = 1$, 2, and 3, respectively[17]. The superscript indicates the spin multiplicity, e.g., $2S + 1 = 1$ for singlet and 3 for triplet, where $S$ is the spin angular momentum. These $L$ terms split due to the direct Coulomb interaction (Fig. 3e). Each $L$ term further splits by the $3p$–$3d$ exchange interaction. The $3p$ and weaker $3d$ spin–orbit interactions remove the degeneracy of the triplet, and mixes up all the $LS$ terms. The spin–orbit interaction is weaker than the exchange interaction, and the effect would be experimentally indistinguishable. We retain the $LS$ notation for the sake of clarity. In addition to the above interactions within an atom, the crystal field also splits the $3d$ level into $e_g$ and $t_{2g}$.

Roughly speaking, the $3p^5 3d^9$ configuration can be categorized energetically into three groups of two terms each, i.e., $^1F$ and $^1P$, $^3D$ and $^3P$, or $^1D$ and $^3F$, which are separated by several electron volts. The two terms in each group happen to have close energies. The intermediate state, $|n\rangle = |1s^1 3d^9\rangle$, is characterized by $1s^1 t_{2g}^6 e_g^3$ and $1s^1 t_{2g}^5 e_g^4$,

because the separation (crystal-field splitting), which was set to 1 eV in the calculation, is larger than a calculated energy splitting of 0.21 eV between $1s^1 3d^9 {}^1D$ and $1s^1 3d^9 {}^3D$ by the exchange interaction. The energy split in $|n\rangle = |1s^1 3d^9\rangle$ is smaller than those in $|i\rangle = |3p^5 3d^9\rangle$. Thus, three satellite peaks in Fig. 3c and 3d correspond to the three groups in $|i\rangle = |3p^5 3d^9\rangle$. This also applies to the Kβ fluorescence and is the origin of a three-peak structure, which has been theoretically predicted but experimentally unresolved[9–12]. The final state, $f$, is dominated by the spin–orbit interaction for the $2p$ orbital. We note that the $1s^1 t_{2g}^6 e_g^3$ and $1s^1 t_{2g}^5 e_g^4$ levels form bands in the real system, and they may be experimentally inseparable in the emission spectrum.

Comparing Fig. 3a with Fig. 3c, we interpret the experimentally resolved clusters as follows. A cluster of strong components (open circles) around (8905 eV, 8048 eV) is the main peak by the $3p^5 \rightarrow 1s^1 \rightarrow 2p^5$ RIXS process. The main peak may be doublet, because the $3p^5$ level splits into $3p_{1/2}$ and $3p_{3/2}$ by the spin–orbit interaction. The separation is reported at about 2.5 eV (ref. 10), and hence, the $3p^5$ ($3p_{1/2}$) related peak may appear around $\omega_p = 8903$ eV, where we found another cluster (open diamonds). The diagonal streak (open triangles and a part of open diamonds) extending from the main peak towards the lower left have similar energy transfers, $\omega_p - \omega_e = 857$ eV. We assign the streak to the tail of the main peak. However, no counterpart was found in the higher photon-energy region. We consider that the components lying along $\omega_e = 8048$ eV apart from the main peak are unseparated satellites and the Kα$_1$ fluorescence from relaxed states.

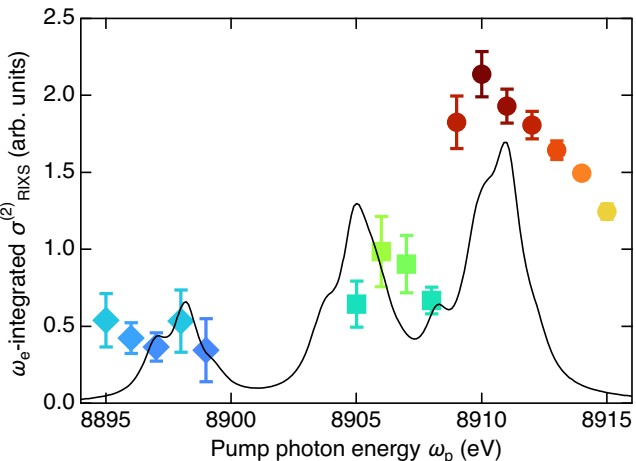

**Fig. 4 | Experimental and theoretical $\omega_p$ dependence of $\omega_e$-integrated $\sigma^{(2)}_{RIXS}$ for $3p^5 3d^9$ satellite.** Symbols represent the peak area of the fitted Lorentzian. Vertical bar indicates the error of the Lorentzian fitting. Diamonds, squares, and circles represent the same components as shown in Fig. 3a. Solid line represents the $\omega_e$-integrated value over the $K\alpha_1$ region of the calculated $\sigma^{(2)}_{RIXS}$ shown in Fig. 3c.

Three clusters labeled B, F, and I (see also Supplementary Fig. 5) are considered to be the satellites involving the $3p^5 3d^9$ $^1F/^1P$, $^3D/^3P$, and $^1D/^3F$ terms, respectively. It is such separation of the $3p^5 3d^9$ satellites from the main peak that we expected for the 2D fluorescence spectrum, and has not been experimentally resolved for the one-dimensional $K\beta$ fluorescence spectrum[9–12]. The theoretical calculation predicted almost the same emission photon energy for these three components, while we find the energies are slightly different. The gravities of the clusters B, F, and I are $\omega_e = 8045.4 \pm 0.06$, $8045.6 \pm 0.11$, and $8046.8 \pm 0.04$ eV, respectively. This implies that transition probabilities to each state in $|n\rangle = |1s^1 3d^9\rangle$ may differ among the $3p^5 3d^9$ $^1F/^1P$, $^3D/^3P$, and $^1D/^3F$ terms in the real system.

The $3p^5 3d^9$ $^1F/^1P$, $^3D/^3P$, and $^1D/^3F$ components less disperse along $\omega_e$, therefore, we discuss the $\omega_p$ dependence by integrating $\sigma^{(2)}_{RIXS}$ over $\omega_e$ within the $K\alpha_1$ region (Fig. 4). The theoretical calculation well reproduces the experimentally deduced spectrum. The agreement is satisfactory, considering that we did not make any empirical adjustments during the calculation. Especially, the energy separations among the three peaks determined mainly by the direct and exchange interactions are in good agreement. However, the relative peak height among the three peaks differs between theory and experiment. This discrepancy may be due to partial extraction of the satellite contribution in our phenomenological analysis of the $K\alpha_1$ emission spectra, and ignored contributions from multi-hole states other than $3p^5 3d^9$.

## Discussion

We successfully realized our nonlinear RIXS scheme on copper metal and measured the 2D fluorescence spectrum as $\sigma^{(2)}_{RIXS}$. Although the main part of the spectrum is simple due to the complete $3d$ subshell in copper metal, we showed that we could isolate the weaker satellite peaks and identify them arising from the two-hole $3p^5 3d^9$ configuration with the help of our multiplet ligand field calculation. Because of the Auger cascade, the initial state of the RIXS part, $i$, is not exactly the same as the final state of the $K\beta$ fluorescence. Nevertheless, $\sigma^{(2)}_{RIXS}$ retained the three-peak satellite structure expected for the $K\beta$ fluorescence spectrum. A possible reason is that solid-state effects could suppress multi-hole states other than the $3p^5 3d^9$ configuration (see Supplementary Note 1).

In the case of compounds including transition metals with incomplete $3d$ subshell, the exchange interaction splits the main peak

into the $K\beta'$ and $K\beta_{1,3}$ peaks, which are known as a useful indicators of the spin and oxidation state[1–3]. For example, the $K\beta'$ peak of Mn(II) compounds is a transition to a $^5P$ final state with the $3p$ spin antiparallel to the five $3d$ spins, while the $K\beta_{1,3}$ peak is to a parallel-spin $^7P$ final state. However, the $K\beta_{1,3}$ peak is also shaped by the $^5P$ term with a different $3d$ spin configuration[29]. The spectrum becomes more complicated for Mn(III) and Mn(IV), which may relate to the oxygen-evolving complex[4–7]. The nonlinear RIXS could serve for a detailed understanding of the spin and oxidation state by resolving such overlapping terms. Our analysis still relies on the phenomenological decomposition of $\sigma^{(2)}_{RIXS}$, which worked relatively well due to the simpler $K\alpha_1$ spectrum for copper metal. However, we anticipate that further theoretical and experimental investigation would lead to sophisticated analysis free from phenomenological modeling.

In the present study, we incorporate the sequential two-photon absorption into RIXS to realize core-to-core absorption. It would be possible to use absorption channels from core to valence[30]. Apart from realizing absorption to the occupied state, other nonlinear processes are also interesting candidates for nonlinear RIXS. Inclusion of direct two-photon absorption is a straightforward application of the present study. Combination of direct two-photon absorption[31–33] and RIXS might be useful to investigate directly the $3d$ state because the $1s \rightarrow 3d$ transition is allowed and the $1s \rightarrow 4p$ is forbidden by the selection rule. Stimulated emission[34–36], which can enhance the fluorescence yield, and select a particular radiative transition of interest, is another interesting nonlinear process to be combined.

## Methods
### Experimental setup
SACLA was operated in self-amplified spontaneous emission (SASE) mode and at a repetition rate of 60 Hz. The half of X-ray pulses were delivered to experimental station 4 of BL3, where we conducted our experiment. The broad-band SASE was tuned to cover the whole $K\beta$ spectral range so that the nonlinear RIXS plane could be measured simply by scanning the monochromator without changing the accelerator and the undulator parameters. The focus size was measured by knife-edge scans with 200-$\mu m\phi$ gold wires. The pulse energy at the focal position was determined shot by shot using a calibrated beam monitor[37]. The shot-by-shot fluctuation was used to reveal the pulse-energy dependence of the emission intensity (Supplementary Fig. 6). The copper foil was continuously shifted to expose the undamaged fresh surface. The foil was tilted by 30° from the normal incidence to measure the X-ray emission at right angles to the incoming beam in the polarization plane (Supplementary Fig. 2). This geometry suppresses stray X-rays due to elastic and Compton scattering.

The single-shot polychromator consisted of four cylindrically bent Si (444) crystals set in the von Hamos geometry[2] and a multi-port charge-coupled device[38] (MPCCD). The crystal dimension was 100 mm along the circumferential direction and 25 mm along the dispersion direction, and the curvature radius was 500 mm. The MPCCD was able to detect a single photon and was suitable to measure the weak nonlinear signal. The nominal photon-energy resolution of the polychromator is calculated to be 0.075 eV/pixel. The pixel number of MPCCD was 1024 along the dispersed direction. The corresponding photon-energy range of the polychromator was about 77 eV, which was enough to cover both the $K\alpha_1$ and $K\alpha_2$ spectral ranges. The absolute photon energy of the polychromator was calibrated to the $K\alpha_1$ peak in literature[10]. The photon energy of the monochromator was calibrated to the absorption edge of a copper foil beforehand.

### Theoretical calculation
Our aim in this study is to calculate the $3d^9$ components of $\sigma^{(2)}_{RIXS}$, i.e., the $3d^9 \rightarrow 3p^5 3d^9 \varepsilon$ and $3p^5 3d^9 \varepsilon \rightarrow 1s^1 3d^9 \varepsilon \rightarrow 2p^5 3d^9 \varepsilon$ RIXS. We used the standard multiplet ligand field theory[16,17,28] with octahedral ($O_h$) crystal field and did not make any empirical adjustments to the term averaged

values of the Slater integrals for the $3d^9$, $3p^53d^9$, $2p^53d^9$, and $1s^13d^9$ configurations. In the spirit of the early phase of the theoretical study, this is a sensible approach and has a more general interest beyond the specific experiment discussed here. The intra-atomic correlation not included in this scheme was treated approximately by reducing the values of the Slater integrals. The Slater integrals and the spin-orbit coupling constants were calculated by the Hartree–Fock code[39], and then the Slater integrals were scaled down to 80% as usual[17]. The crystal field describes the breaking of the $3d$ orbital degeneracy in copper metal due to the effect of the electrical field of neighboring copper atoms. The crystal-field splitting parameter was set to $10Dq = 1.0$ eV. We assumed that the shake-up process is negligible[12], and the shake-off probability does not strongly depend on the details of the multiplet. We calculated $\sigma_{gi}$ in Eq. (1) as a $3d^9 \rightarrow 3p^53d^9\varepsilon$ transition process instead of $3d^{10} \rightarrow 3p^53d^9\varepsilon^2$ (ref. 10). Because of a similar reason, we set $\tau_i = 0.15$ eV, however, it may strongly depend on the spin state in other $3d$ configurations[28].

## Data availability
The datasets that support the results of this study can be found in this published article. Source data are provided with this paper.

## Code availability
The codes of the computer simulations are available from M.T. upon request.

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

## Acknowledgements

This work was supported by a grant-in-aid from the Ministry of Education, Culture, Sports, Science and Technology of Japan under grant number 21H03746 (K.T.). The experiments were performed with the approval of JASRI under proposal numbers 2017B8034, 2018A8046, 2018B8057, 2019A8049, 2020A8016, and 2021A8018 (K.T.).

## Author contributions

K.T. conceived the study. T.O., I.I., Y.I., and K.T. acquired the experimental data. M.T. performed multiplet ligand field calculation. K.T. analyzed data. M.T. and K.T. wrote the manuscript. T.I. and M.Y. supervised the work. All authors contributed to discussion and manuscript revisions.

## Competing interests

The authors declare no competing interests.
