## [Peer Review File · Nature Communications]

Two-dimensional $K\beta$ - $K\alpha$ fluorescence spectrum by nonlinear resonant inelastic X-ray scattering

Editorial Note: Parts of this Peer Review File have been redacted as indicated to remove third-party material where no permission to publish could be obtained.REVIEWER COMMENTS

Reviewer #1 (Remarks to the Author):

The data of the Two-dimensional Kb-Ka fluorescence spectra are very interesting and worth publishing.

Before the manuscript can be published, I have some questions on the theoretical modelling.

(1) In the first step a 3p core hole is created with a high energy x-ray. This is essentially the 3p XPS (HAXPES) spectrum of Cu metal, which in first approximation consists of two peaks split by the 3p spin-orbit coupling of ~ 2.5 eV.

(2) I note that the Ka and Kb fluorescence spectra should be calculated from the convolution of the 1s XPS spectrum and the 1s3d XES spectrum, as was nicely shown by Glatzel et al. for different excitation processes [Phys Rev B 64, 045109].

Following this analysis the 1s3p x-ray absorption step starts from a 3p XPS created state, which contains two peaks. Assuming a final state with a single 1s core hole related peaks this yields two peaks, respectively at ~ 8905 and ~ 8902.5 eV.

(3) Based on this analysis I interpret figure 2 in a different way:

8897 and 8912 are not at a 1s3p resonance thus one observes the Raman tail of the two resonances (essentially the 3p XPS spectrum), respectively B and C; I and J.

8905 is exactly at the 3p_{3/2} resonance and the 3p_{1/2} peak is suppressed.

(The 1s2p XES step does add the 2p spin-orbit coupling but (in first approximation) nothing else).

(4) A second issue is the following: The system is Cu metal that has a 3d¹⁰ ground state. How can core hole creation ever create a 3d⁹ state? The 3p core hole will be screened by 3d electrons, so the ground state will remain (at least 99%) 3d¹⁰. The same applies to a 1s and 2p core hole state. Could the authors indicate why they assume a visible signal related to a 3d⁹ state?

Signed: Frank de Groot

Reviewer #2 (Remarks to the Author):

The Authors present an interesting realization of two-photon scattering process at $K\beta$ resonance, that leads to enhancement of signal from two-hole satellite states in Cu metal, hardly visible in conventional fluorescence spectra. The work is a valuable contribution in exploring the potential for non-resonant x-ray emission studies. Non-linear x-ray studies, enabled by free-electron laser facilities, are still rare.

The overall study is sound and the conclusions are well thought out. The experimental observations are nicely combined with model calculations. Nevertheless, some details on the data analysis could be specified better.

In particular, following points could be specified/improved:

- 1) One important point to clarify is what are the intensity maps in Fig. 1 c and d exactly showing. Is this the actually measured fluorescence intensity (e.g. integrated over different pulse energies?), or is it the coefficient of the quadratic term after analysis obtained for each data point as shown Fig. 1b? (Last could be understood from last sentence of 2nd paragraph on p. 6.)
- 2) It seems to me that there is some inconsistency in referring to the data through the manuscript. While some phrases in the manuscript (see "Introduction" and "Discussion") are describing the data as "two-dimensional $K\alpha$ - $K\beta$ fluorescence", in other parts the data is described as non-linear RIXS.
- 3) It would be valuable to add a histogram of the pulse energies in Supplementary Information (in reference to e.g. Fig. 2b).
- 4) When discussing the fit results in last paragraph of section "Nonlinear RIXS spectrum", it would be helpful to reference Fig. S2 (Supplementary Information). For this figure a larger displacement in the vertical direction would help to see more details in the spectra (and also the fitted peaks). At the same time, a guide to the eye for the Raman-like structure (G') and also for the structure at constant emission energy would illustrate better their energy dependence.
- 5) In Fig. 3a and 3b there are missing explanations of some symbols (in Fig. 3a the open triangles and circles are not described, while in Fig. 3b the description is missing).
- 6) I did not understand, whether $K\beta$ fluorescence was also measured in this study. The sentence on p. 6 says: "The emission spectrum at $\omega_p=8905$ eV, i.e., the $K\beta$ peak, is found similar to the $K\alpha$ fluorescence spectrum." If this was true, it would be valuable to include the data in the Supplementary Information.
- 7) The Authors make the statement that an absorption process was equivalent to a fluorescence process (between the same states). It is not generally true, since they both probe different density of states – therefore, it would be good to be more specific here (section "Nonlinear RIXS scheme").
- 8) In the Discussion the Authors say that non-linear $K\beta$ fluorescence from systems with partially occupied 3d states could reveal $K\beta$ superstructure without the contribution of the satellites. Why would

this be the case, in contrast to systems with fully occupied 3d states (and which satellites would not be visible)?

Reviewer #3 (Remarks to the Author):

The present study by Tamasaku et al proposes a new scheme for studying atomic transitions and electronic states using a nonlinear RIXS scheme. The proposal is definitely novel, rather interesting, and peculiar. It might have some potential for future development in the field of atomic spectroscopy. However, in my opinion, it is beyond the general scientific interest of the broad Nature Communication audience. I may suggest submitting the paper to a more specific physical journal, such as Physical Review Letters, as this new observation definitely deserves the attention of the devoted community of atomic spectroscopy. Besides that, the paper in its present form shows several weaknesses and raises a few questions which I summarised below and may recommend authors to attend those points in the next manuscript edition.

- 1) Introduction would benefit of citing some new reviews on resonant X-ray scattering techniques, e.g. Gelmukhanov et al, Rev. Mod. Phys. (2021).
- 2) Authors mentioned the possibility of description of electronic processes in the high X-ray field using the rate equations. I think it must be done for a good publication. Without this there is an open question concerning how many other channels can be opened by high intensity X rays and how much they may affect the studied processes? E.g. one- and multy-photon ionisation, cascade Auger, excitation of ionic states, etc. At least, those possible channels must be discussed.
- 3) Authors wrote explicit equations for the cross section, yet not giving the units in which they are given.
- 4) In equations (1-2) how many intermediate (i) and final (f) states were considered? In that sense I see a serious contradiction between the rather simple schematic representation of Fig. 3(e) and numerous fitted Gaussians of Fig. 3(a-b).
- 5) The two peaks of actual experimental data of Fig. 1(c) are structureless. The authors managed to make a fitting with 3 Gaussian functions in Fig. 2, yet they fitted them with numerous gaussians in Fig. 3. The question arises: how can authors be sure that this fitting reflects the real electronic structure? If the deconvolution is only possible with the help of numerical simulations of the electronic structure, the experimental method itself cannot be considered as a prominent tool.

6) Due to the above mentioned, even analysis of an atomic system (Cu atom) with the present methodology looks quite uncertain to me. Could authors comment on possible application of the proposed scheme for studies of more advanced systems, such as metallo-organic complexes, etc? I believe, if that link can be clearly shown, the method can attract much attention in the future, given that spectral resolution may improve significantly for the coming X-ray instruments.

Response to Reviewer #1:

The data of the Two-dimensional K α -K β fluorescence spectra are very interesting and worth publishing.

Before the manuscript can be published, I have some questions on the theoretical modelling.

We thank the Reviewer #1 for the careful reading of our manuscript and the positive evaluation.

(1) In the first step a 3p core hole is created with a high energy x-ray. This is essentially the 3p XPS (HAXPES) spectrum of Cu metal, which in first approximation consists of two peaks split by the 3p spin-orbit coupling of ~2.5 eV.

(2) I note that the K α and K β fluorescence spectra should be calculated from the convolution of the 1s XPS spectrum and the 1s3d XES spectrum, as was nicely shown by Glatzel et al. for different excitation processes [Phys Rev B 64, 045109].

Following this analysis the 1s3p x-ray absorption step starts from a 3p XPS created state, which contains two peaks. Assuming a final state with a single 1s core hole related peaks this yields two peaks, respectively at ~8905 and ~8902.5 eV.

(3) Based on this analysis I interpret figure 2 in a different way:

8897 and 8912 are not at a 1s3p resonance thus one observes the Raman tail of the two resonances (essentially the 3p XPS spectrum), respectively B and C; I and J.

8905 is exactly at the 3p $_{3/2}$ resonance and the 3p $_{1/2}$ peak is suppressed.

(The 1s2p XES step does add the 2p spin-orbit coupling but (in first approximation) nothing else).

We believe that we observed two contributions, the main peak, and the 3d-related satellites. The main peak is the 3p $^5 \rightarrow 1s^1 \rightarrow 2p^5$ transition (Fig.1a) at 8905 eV, where the 3p 5 state is created by the 3p photoionization by the first pump photon, as pointed out in the comments 1 and 2. However, we cannot experimentally resolve the doublet, i.e., 3p $_{1/2}$ and 3p $_{3/2}$. The 3d-related satellites come from the 3p $^5 3d^9 \rightarrow 1s^1 3d^9 \rightarrow 2p^5 3d^9$ transition. The reason why the initial photoionization process creates the 3p $^5 3d^9$ state is explained in the next reply.

We consider that it is difficult to interpret the peaks B, C, I, and J in Fig.2 as the Raman tail of the main peak. In the figure below, we indicate the positions for these peaks on the same $\omega_p - \omega_e$ plot as Fig. 3a. The Raman tail should lie near the diagonal dashed line, which indicates a constant energy transfer line passing through the main peak. Note that the vertical axis of the figures is not the energy transfer but the emission photon energy. The peaks B, C, I, and J are far from this line, which implies origins other than the Raman tail.

(4) A second issue is the following: The system is Cu metal that has a 3d¹⁰ ground state. How can core hole creation ever create a 3d⁹ state? The 3p core hole will be screened by 3d electrons, so the ground state will remain (at least 99%) 3d¹⁰. The same applies to a 1s and 2p core hole state. Could the authors indicate why they assume a visible signal related to a 3d⁹ state?

The ground state may be expressed as $3d^{10}+3d^9L$ with the dominant $3d^{10}$ state in the charge-transfer multiplet theory, while it is $3d^{10}$ in the multiplet ligand theory employed in this study. So, the ground state has no $3d^9$ state in our theoretical model. However, the excited state, i , contains the $3p^53d^9$ state, due to the shake-off process during the first non-resonant absorption. This is not specific to nonlinear RIXS, and the shake-off process is considered to produce the $3p^53d^9$ satellites in the $K\beta$ fluorescence spectrum (Refs. 9-12).

In the revised manuscript, we added another process, the Auger cascade, which may create the $3p^53d^9$ state as pointed out by the Reviewer #3. To make clear these points, i.e., the shake-off and Auger processes, we revised the text at p. 4, lines 13-20.

Response to Reviewer #2:

The Authors present an interesting realization of two-photon scattering process at $K\beta$ resonance, that leads to enhancement of signal from two-hole satellite states in Cu metal, hardly visible in conventional fluorescence spectra. The work is a valuable contribution in exploring the potential for non-resonant x-ray emission studies. Non-linear x-ray studies, enabled by free-electron laser facilities, are still rare.

The overall study is sound and the conclusions are well thought out. The experimental observations are nicely combined with model calculations. Nevertheless, some details on the data analysis could be specified better.

We thank the Reviewer #2 for the careful reading of the manuscript, and the positive and the constructive comments below. The revisions made in accordance with the comments greatly improved readability and accessibility of our manuscript.

1) One important point to clarify is what are the intensity maps in Fig. 1 c and d exactly showing. Is this the actually measured fluorescence intensity (e.g. integrated over different pulse energies?), or is it the coefficient of the quadratic term after analysis obtained for each data point as shown Fig. 1b? (Last could be understood from last sentence of 2nd paragraph on p. 6.)

Figures 1c and 1d show $\sigma_{\text{#}}^{(\xi_{\text{#}})}$ which is the quadratic coefficient. To clarify this point, we revised the caption of Figs.1 and 4, and the vertical label of Fig. 4 and Supplementary Fig. 4.

2) It seems to me that there is some inconsistency in referring to the data through the manuscript. While some phrases in the manuscript (see "Introduction" and "Discussion") are describing the data as "two-dimensional $K\alpha - K\beta$ fluorescence", in other parts the data is described as non-linear RIXS.

We omitted the phrase "nonlinear RIXS spectrum" from the revised manuscript. We used $\sigma_{\text{#}}^{(\xi_{\text{#}})}$ to indicate the measured and calculated spectra, and "2D fluorescence spectrum" when its physical meaning is important. In this connection, we changed the subsection headings in Results, and divided "Nonlinear RIXS spectrum" subsection of the original manuscript into two parts, "Measurement of $\sigma_{\text{#}}^{(\xi_{\text{#}})}$ " and "2D fluorescence spectrum".

3) It would be valuable to add a histogram of the pulse energies in Supplementary Information (in reference to e.g. Fig. 2b).

We added Supplementary Fig. 6, which shows the histogram of the pulse energies during the measurement of Fig. 1b. A reference to Supplementary Fig. 6 was added in Methods, because there is a related explanation about how to measure the pulse-energy dependence.

4) When discussing the fit results in last paragraph of section “Nonlinear RIXS spectrum”, it would be helpful to reference Fig. S2 (Supplementary Information). For this figure a larger displacement in the vertical direction would help to see more details in the spectra (and also the fitted peaks). At the same time, a guide to the eye for the Raman-like structure (G') and also for the structure at constant emission energy would illustrate better their energy dependence.

We added a reference to Supplementary Fig. 4 (Supplementary Fig. 2 in the original manuscript) at the beginning of “Analysis of $\sigma_{\text{R}}^{(\text{S})}$ ” subsection (p. 8, line 10), where we elaborated the Lorentzian fitting. We also revised Supplementary Fig. 4 according to the comment.

5) In Fig. 3a and 3b there are missing explanations of some symbols (in Fig. 3a the open triangles and circles are not described, while in Fig. 3b the description is missing).

We revised the caption of Figs. 3a and 3b.

6) I did not understand, whether $K\beta$ fluorescence was also measured in this study. The sentence on p. 6 says: “The emission spectrum at $\omega_p=8905$ eV, i.e., the $K\beta$ peak, is found similar to the $K\alpha$ fluorescence spectrum.” If this was true, it would be valuable to include the data in the Supplementary Information.

We have measured the $K\beta$ fluorescence spectrum, which was added as Supplementary Fig. 3.

7) The Authors make the statement that an absorption process was equivalent to a fluorescence process (between the same states). It is not generally true, since they both probe different density of states – therefore, it would be good to be more specific here (section “Nonlinear RIXS scheme”).

We revised the sentence, which is “... govern by the same transition matrix element” in the revised manuscript (p. 3, lines 16-17).

8) In the Discussion the Authors say that non-linear $K\beta$ fluorescence from systems with partially occupied 3d states could reveal $K\beta$ superstructure without the contribution of the satellites. Why

would this be the case, in contrast to systems with fully occupied 3d states (and which satellites would not be visible)?

We revised the discussion about the incomplete 3d system (p. 11, lines 12-20) according to the comments from Reviewer #2 and #3, where we picked up a Mn 3d⁵ system. We added a related reference (Ref. 29).

Response to Reviewer #3:

The present study by Tamasaku et al proposes a new scheme for studying atomic transitions and electronic states using a nonlinear RIXS scheme. The proposal is definitely novel, rather interesting, and picular. It might have some potential for future development in the field of atomic spectroscopy. However, in my opinion, it is beyond the general scientific interest of the broad Nature Communication audience. I may suggest submitting the paper to a more specific physical journal, such as Physical Review Letters, as this new observation definitely deserves the attention of the devoted community of atomic spectroscopy. Besides that, the paper in its present form shows several weaknesses and raises a few questions which I summarised below and may recommend authors to attend those points in the next manuscript edition.

We thank the Reviewer #3 for the careful reading of the manuscript, and giving several important comments below, which deepened our understanding of the nonlinear RIXS process, strengthened the logic, and made the manuscript comprehensible. We believe that the revised manuscript can attract wider audience of Nature Communications.

1) Introduction would benefit of citing some new reviews on resonant X-ray scattering techniques, e.g. Gelmukhanov et al, Rev. Mod. Phys. (2021).

We added the reference as Ref. 18.

2) Authors mentioned the possibility of description of electronic processes in the high X-ray field using the rate equations. I think it must be done for a good publication. Without this there is an open question concerning how many other channels can be opened by high intensity X rays and how much they may affect the studied processes? E.g. one- and multy-photon ionisation, cascade Auger, excitation of ionic states, etc. At least, those possible channels must be discussed.

We used rate equations to describe the sequential two-photon absorption part in the nonlinear RIXS process, instead of treating it as a single coherent three-photon process. When the rate equations are solved under an appropriate approximation, the cross section may be expressed in a form of $\sigma_{gl}\tau_i\sigma_{in}$ (Ref. 23). However, σ_{in} should be replaced with σ_{RIXS} in the present case. We do not intend to point out the rate-equation approach to handle a complex electronic process invoked by high-intensity X-rays. To avoid confusion, we revised the sentence at p. 4, lines 24-25.

The multi-photon ionization, which involves more than two photons, is negligible, as is experimentally verified by the pulse-energy dependence of the emission reproduced well by a quadratic term only (Fig. 1b and p. 6, lines 12-15).

As for Auger cascade, we simulated the Auger cascade using coupled rate equations and an atomic model, however, the results failed to explain the experimental results. We consider that the discrepancy is due to some states having longer lifetime in the atomic model, which can decay quickly in a solid-state environment. We couldn't incorporate such solid-state effects in our simulation, so that we added a qualitative discussion about the reason why the measured spectra can be explained by the $3p^5$ main peak and the $3p^53d^9$ satellites in Discussion at p. 11, lines 7-11 and Supplementary Note 1. We also revised the second paragraph on p.4 to include the contribution of the Auger cascade (p. 4, lines 13-20), and the text at p. 10, lines 24-26.

3) Authors wrote explicit equations for the cross section, yet not giving the units in which they are given.

We added a sentence describing the dimension of $\sigma^{(i,f)}$ at p. 5, lines 7-9. We found equation (2) was incorrectly typed, and revised it. The simulation was performed using correct formula, and is not affected by this correction.

4) In equations (1-2) how many intermediate (i) and final (f) states were considered? In that sense I see a serious contradiction between the rather simple schematic representation of Fig. 3(e) and numerous fitted Gaussians of Fig. 3(a-b).

The number of states used in the numerical simulation was 60 for i and f , and 20 for n . In Fig.3e, we did not show detailed fine structures, which cannot be experimentally resolved. The number of Lorentzians used for fitting (Figs. 3a and 3b) does not indicate the number of the spectral components. We sliced the 2D spectrum along the constant ω_p line, and fitted each slice (Supplementary Fig. 4). Therefore, one spectral feature may be divided into several Lorentzians (Supplementary Fig. 5).

Because this is the key point of our analysis free from prior knowledge, we moved the description about the fitting from Methods to the main text at p. 8, lines 8-13, and added an explanatory sentence there and Supplementary Fig. 5.

5) The two peaks of actual experimental data of Fig. 1(c) are structureless. The authors managed to make a fitting with 3 Gaussian functions in Fig. 2, yet they fitted them with numerous gaussians in Fig. 3. The question arises: how can authors be sure that this fitting reflects the real electronic

structure? If the deconvolution is only possible with the help of numerical simulations of the electronic structure, the experimental method itself cannot be considered as a prominent tool.

We can see some structures even in the image plot, but to make them clearer we added contour lines in Fig.1c. The reason why we had many Lorentzians is explained in our response to the comment 4 above. In fact, the number of resolved spectral feature was only five, which are the $3p^5$ main peak, its Raman tail, and three $3p^5 3d^9$ satellites. Throughout our analysis of the measured $\sigma_{\#}(\xi_{\#})$, we did not need help of the numerical simulation, and used it to assign the LS terms to the three satellites. We would like to note that one need to consult with theory to interpret the $K\beta$ fluorescence spectrum. In addition, the analysis of the 2D spectrum will not need a full simulation of the nonlinear RIXS process as we have shown in the present work; the level diagram for the resonant absorption part or equivalently the $K\beta$ fluorescence is sufficient to interpret the 2D fluorescence spectrum.

6) Due to the above mentioned, even analysis of an atomic system (Cu atom) with the present methodology looks quite uncertain to me. Could authors comment on possible application of the proposed scheme for studies of more advanced systems, such as metallo-organic complexes, etc? I believe, if that link can be clearly shown, the method can attract much attention in the future, given that spectral resolution may improve significantly for the coming X-ray instruments.

Firstly, we would like to emphasize that the $3p^5 3d^9$ satellites have not been isolated until the present study, though they had been theoretically predicted in late 1970s (Ref. 9), and that we analyzed the spectrum under a minimum assumption without help of theory (the 1st paragraph of “Analysis of $\sigma_{\#}(\xi_{\#})$ ” section). We would also like to note that the theoretical framework we used here is not a simple atomic model but multiplet ligand field theory. In this connection, we replaced the phrase “atomic multiplet theory” in the original manuscript with “multiplet ligand theory”.

We elaborated a possible application to the Mn system at p. 11, lines 12-20, which is directly related to the oxygen-evolving complex of photosystem II (PSII). We expect that decomposition of overlapping terms by nonlinear RIXS would greatly help to understand a subtle change in the Mn oxidation state during the water-splitting cycle of PSII.

REVIEWER COMMENTS

Reviewer #1 (Remarks to the Author):

See the attached file & figures

I thank the authors for the explanations. I agree with the analysis of the satellite peaks in normal Kbeta x-ray emission spectra.

Concerning the 3p XPS excitation the authors write “we cannot experimentally resolve the doublet, i.e., $3p_{1/2}$ and $3p_{3/2}$.” This is a bit strange as the splitting is very clear, as shown in a spectrum from literature, copied here:

[REDACTED]

Starting from the main peak the transitions in the experiment are:

GROUND > $3p_{3/2}$ (75 eV) > $1s$ (+8905 eV) > $2p$ (-8048 eV)

This yields the main peak when the system is excited at 8905 eV. I do not know how to exactly interpret the symbols in the figure below that was given in the rebuttal, but I would assign the triangles and the green square symbols at 8046 eV as the $3p_{1/2}$ shoulder via the process:

GROUND > $3p_{1/2}$ (77 eV) > $1s$ (+8903 eV) > $2p$ (-8046 eV)

Both these $3p_{3/2}$ and $3p_{1/2}$ channels will then have the satellites from “shake processes” and Auger as described in the paper.

Do the authors agree with the possibility of this assignment?
Then it would be useful to describe it as option.

Reviewer #2 (Remarks to the Author):

I would like to thank the Authors for addressing all my comments. After reading the revised version, I would still like to raise these points:

1) The experimentally observed scattering intensities are usually related to differential cross sections – as one would also expect for the coefficient of the quadratic term in the two-photon scattering process. However, equation (1), which is used to simulate $\sigma^{(2)}_{\text{RIXS}}$, seems to be expressed as total integrated cross section. Then in Fig. 4, this quantity is integrated. Please, check the formalism of the equations. At the same time, equation (2) is expressed as double-differential cross section.

2) On p. 5, 2nd paragraph from the bottom there is a sentence that reads as general statement that conventional RIXS was probing mainly the intermediate state. This is not true, since for many studies the ground and final state are at least equally important.

Reviewer #3 (Remarks to the Author):

Authors addressed carefully all remarks, making manuscript more comprehensive. In my opinion, the paper now can be accepted for publication.

Response to Reviewer #1:

I thank the authors for the explanations. I agree with the analysis of the satellite peaks in normal Kbeta x-ray emission spectra.

We thank the Reviewer #1 again for reading the revised manuscript, and for very interesting comment on interpretation of the nonlinear RIXS spectrum. According to the comment, we revised our manuscript as follows.

Concerning the 3p XPS excitation the authors write “we cannot experimentally resolve the doublet, i.e., $3p_{1/2}$ and $3p_{3/2}$.” This is a bit strange as the splitting is very clear, as shown in a spectrum from literature, copied here:

[REDACTED]

Starting from the main peak the transitions in the experiment are:

GROUND > $3p_{3/2}$ (75 eV) > 1s (+8905 eV) > 2p (-8048 eV)

This yields the main peak when the system is excited at 8905 eV. I do not know how to exactly interpret the symbols in the figure below that was given in the rebuttal, but I would assign the triangles and the green square symbols at 8046 eV as the $3p_{1/2}$ shoulder via the process:

GROUND > $3p_{1/2}$ (77 eV) > 1s (+8903 eV) > 2p (-8046 eV)

Both these $3p_{3/2}$ and $3p_{1/2}$ channels will then have the satellites from “shake processes” and Auger as described in the paper.

Do the authors agree with the possibility of this assignment?

Then it would be useful to describe it as option.

Firstly, we had meant that we could not resolve the doublet structure by the $3p_{1/2}$ - $3p_{3/2}$ splitting in the nonlinear RIXS spectrum. This is because we had considered that the $3p_{1/2}$ -related peak

should have the same emission photon energy, ω_e , as the $3p_{3/2}$ -related peak, and that such overlapped peaks could not be resolved due to the reason discussed in Introduction. However, the $3d$ -related satellite peaks were found to have different ω_e (p.10, lines 19-20), and it's likely that the doublet is separated vertically in the ω_p - ω_e plane. Therefore, we would agree with the interpretation proposed by the Reviewer #1, while we kept our assignment of Cluster F unchanged, because the peak F in Fig.2 should be the satellite (discussion in the second paragraph in p.7). We revised the symbols in Fig. 3a to distinguish the $3p_{1/2}$ - and $3p_{3/2}$ -related components, and the caption. We added two sentences and modified the next one at p.10, lines 3-8 to explain the new interpretation. We note that the new interpretation does not affect the discussion and the conclusion, where we focused on the satellites.

Response to Reviewer #2:

I would like to thank the Authors for addressing all my comments. After reading the revised version, I would still like to raise these points:

We thank the Reviewer #2 again for reviewing our manuscript, and for careful checking of the formula. According to the comments, we revised our manuscript as follows.

1) The experimentally observed scattering intensities are usually related to differential cross sections – as one would also expect for the coefficient of the quadratic term in the two-photon scattering process. However, equation (1), which is used to simulate $\sigma^{(2)}_{\text{RIXS}}$, seems to be expressed as total integrated cross section. Then in Fig. 4, this quantity is integrated. Please, check the formalism of the equations. At the same time, equation (2) is expressed as double-differential cross section.

We corrected equation (1), which describes the relation between the differential cross section and the double differential cross section given by equation (2). We also added explanation on new variables and function in the text (p.5, lines 3-15).

We would like to use $\sigma^{(2)}_{\text{RIXS}}$ in the rest of the manuscript for simplicity, because $\sigma^{(2)}_{\text{RIXS}}$ is proportional to $d\sigma^{(2)}_{\text{RIXS}}/d\Omega_e$. The numerical simulation based on equation (2) does not depend on the emission direction. The experiment was performed for polycrystalline copper foils, and the measured data was averaged over crystal orientations, even if there was any directional dependence.

The ω_e -integration in Fig.4 is performed for $\sigma^{(2)}_{\text{RIXS}}$ (not $d^1\sigma^{(2)}_{\text{RIXS}}/d\hbar\omega_e d\Omega_e$) to make the decomposed spectrum one-dimensional. After decomposition, the information along the ω_e direction may be unimportant.

2) On p. 5, 2nd paragraph from the bottom there is a sentence that reads as general statement that conventional RIXS was probing mainly the intermediate state. This is not true, since for many studies the ground and final state are at least equally important.

We agree that the original sentence was misleading, and rewrote it (p.5 lines 18-20) to point out that the initial state of the RIXS part is not the ground state but includes many excited states with different energies.

Response to Reviewer #3:

Authors addressed carefully all remarks, making manuscript more comprehensive. In my opinion, the paper now can be accepted for publication.

We thank the Reviewer #3 for reviewing our manuscript, and recommending it for publication.

REVIEWERS' COMMENTS

Reviewer #1 (Remarks to the Author):

I agree with the modifications.

I suggest publication of the manuscript.

Reviewer #2 (Remarks to the Author):

The Authors addressed all my comments satisfactory. I do not have further remarks.

Response to Reviewer #1:

I agree with the modifications.

I suggest publication of the manuscript.

We appreciate the Reviewer #1 for positive feedback and supporting publication.

Response to Reviewer #2:

The Authors addressed all my comments satisfactory. I do not have further remarks.

We appreciate the Reviewer #2 for positive feedback.